# Transmission-Based Vertebrae Strength Probe Development: Far Field Probe Property Extraction and Integrated Machine Vision Distance Validation Experiments

**DOI:** 10.3390/s23104819

**Published:** 2023-05-17

**Authors:** Paul Meaney, Robin Augustine, Adrian Welteke, Bernd Pfrommer, Adam M. Pearson, Helena Brisby

**Affiliations:** 1Thayer School of Engineering, Dartmouth College, Hanover, NH 03755, USA; 2Electrical Engineering Department, Uppsala University, 751 05 Uppsala, Sweden; 3Electrical Engineering Department, Helmut Schmidt University, 22043 Hamburg, Germany; 4Independent Researcher, Hudson, FL 34667, USA; 5Geisel School of Medicine, Dartmouth College, Lebanon, NH 03766, USA; 6Orthopedic Department, Sahlgrenska Hospital, 413 45 Gothenburg, Sweden

**Keywords:** vertebrae, instrumentation, osteoporosis, microwave, transmission, machine vision, surgical navigation

## Abstract

We are developing a transmission-based probe for point-of-care assessment of vertebrae strength needed for fabricating the instrumentation used in supporting the spinal column during spinal fusion surgery. The device is based on a transmission probe whereby thin coaxial probes are inserted into the small canals through the pedicles and into the vertebrae, and a broad band signal is transmitted from one probe to the other across the bone tissue. Simultaneously, a machine vision scheme has been developed to measure the separation distance between the probe tips while they are inserted into the vertebrae. The latter technique includes a small camera mounted to the handle of one probe and associated fiducials printed on the other. Machine vision techniques make it possible to track the location of the fiducial-based probe tip and compare it to the fixed coordinate location of the camera-based probe tip. The combination of the two methods allows for straightforward calculation of tissue characteristics by exploiting the antenna far field approximation. Validation tests of the two concepts are presented as a precursor to clinical prototype development.

## 1. Introduction

There are roughly 900,000 spinal fusion surgeries in the US each year [1,2,3]. In most cases, an instrumentation is fabricated to hold the spine rigid for several months to allow the neighboring vertebrae to fuse [4]. These instruments generally consist of two rods positioned along the spinal column which are held in place by sets of brackets with associated screws proceeding through the pedicles and into the large part of each vertebra [4,5]. On occasion there are instrumentation failures, sometimes with catastrophic consequences, and many are associated with weak bones that are insufficiently strong to withstand the strain of the screws [6,7]. A high percentage of these failures are related to osteoporosis [8]. To minimize the potential for failures, presurgical dual energy X-ray absorptiometry (DXA) scans are performed to assess the health of the vertebrae [9,10]. DXA scans are most useful in deriving a systemic assessment of bone health but can be problematic in examining the spine [10]. Complications can include technician variability, variable density between adjacent vertebrae and the presence of microcalcifications in the overlying tissue [8]. More importantly, DXA scans are nearly useless when examining the thoracic region of the spine because of the presence of the overlying ribcage [10]. Alternatives to DXA include MRI which has been used to provide an MRI-based vertebral bone quality score [11,12].

Determining vertebrae strength is critical to the design of the instrumentation [10]. In the event that the bones are weaker than normal, steps can be taken to alleviate problems [9,10]. These include incorporating more vertebrae to spread out the strain, using anatomical cements to increase the effective bone strength and using specialized screws [13,14]. Each technique has advantages and disadvantages, but the crucial factor is the need for quick and reliable assessments of bone strength. For this, surgeons need a point-of-care sensor for quickly and reliably assessing the vertebrae strength at the time of surgery [13]. Exploiting the specificity embedded in the tissue dielectric properties, is one promising way to evaluate the bone [15,16]. Before fabricating the instrumentation, the surgeon makes small screw guide holes into each vertebra through the pedicles on both sides [6,13]. These are ideal for inserting a thin probe and taking a measurement. Commercial, reflection-based dielectric probes are appealing but inadequate for this application because the penetration depth of the probe is so shallow, and the associated calibration is dependent on the probe remaining stationary [17,18,19]. Optical techniques could be attractive in the sense that it would be relatively easy to insert a fiber optic cable into the screw holes. However, the light attenuation in bone for even small distances would be prohibitive [20]. A microwave transmission-based coaxial probe is feasible for this because the hole pairs are created so that the probes could be positioned on opposing sides of the bone without any additional invasive steps [21].

The key insight for our dielectric probe is that it operates in transmission mode compared with more conventional reflection-based probes [21]. In fact, a major precedent for the transmission concept in deriving material and tissue properties is the vast amount of testing performed in the ultrasound arena [22]. As noted earlier, there are a series of critical features which make this concept feasible for actual clinical use, whereas the reflection-based system will most likely continue to be relegated to bench-top, laboratory scale experiments [23]. Primarily, the conventional probes require the probe and cable to be fixed in position. Any perturbation of the cables or probe position can easily negate the calibration and render it useless unless repeatedly being refreshed against known material standards. The transmission probe has proven to be remarkably resilient regarding cable motion and is well suited for applications where the probes need to be continually re-positioned. Interestingly, the coaxial apertures are sufficiently small that the far field radiation pattern begins roughly 1 mm or closer for the entire operating frequency band [24]. This implies that the phase and amplitude vary linearly as a function of distance from each probe [25], dramatically simplifying the calculation of the dielectric properties from the measurements [21].

More importantly, the transmission probes can interrogate a considerably larger sample volume, thus providing a weighted average property calculation that is less vulnerable to artifacts from poor probe/sample contact since the interface space comprises only a very small fraction of the interrogation volume [17,18]. The commercially available reflection-based probes only have sensing volumes on the order of 0.3 mm deep which make them prone to problems from inadequate contact [17,18,19]. This is particularly problematic for bone which normally has to be manipulated more in terms of cutting and sawing than other soft tissue to make a flat surface for the probe [26]. Additionally, for measuring properties deep within a hole, there is considerable uncertainty regarding the probe/tissue contact interface quality. The transmission probe is limited by how much tissue the signal can propagate across. From our experience, using a vector network analyzer (VNA) with a dynamic range on the order of 140 dB, we can readily measure samples as large as 4 cm thick. This provides a far superior rendering of the overall tissue content than a mere superficial sampling—even when the probe contact is unsure.

Similar to the conventional probes, our new probes can operate over a very large bandwidth—typically from about 1–8 GHz. In spite of the fact that the impedance presented at the ends of each coax is very nearly an open circuit across a wide band, that impedance is applied very uniformly for the full band. While this implies that a significant majority of the fields are reflected back at the aperture interface, the spectral transmission is remarkably flat. Our experience is that the probe combination loses roughly 40 dB of signal due to these discontinuities; however, with a measurement system dynamic range of roughly 140 dB, there is more than enough signal for tissue evaluation.

The probe handles consist of a simple semi-rigid coaxial cable running the full length of the 3D printed plastic handles with a right-angle bend and extra coaxial adapter so that the connector can protrude out the handle sides for connection to cables from the vector network analyzer (VNA). For the distance measurement feature, we exploit a machine vision concept that is widely used in visual product inspection based on deep learning methods and collaborative assembly task realization using selected types of human–robot interaction [27,28]. We configure one handle with a pair of printed tags of known identity and size—printed on paper and glued to a smooth 1 mm thick sheet of fiberglass. Attached to the second handle is a small web camera which continuously records the tags and feeds the images to a computer via a USB link (Figure 1). Before deploying the probes, we use the calibration procedure described in Section 2.1 to determine the camera’s projective properties and the full probe geometry, i.e., the static positions of the probe tip with respect to the tags or the camera, respectively. When the probes are in use, a real-time computer vision algorithm extracts the tag corner identities and locations from the image stream and leverages the known projective camera properties to compute the missing non-static geometric transform between the tags on one handle and the camera on the other handle thus enabling the calculation of the probe tip separation distance.

Section 2 describes the probe fabrication, use and associated calibrations for the machine vision separation distance operation. It also includes a description of a surgical navigation-based implementation which is used for validation testing of the former. It briefly summarizes the electromagnetic operation of the probes—especially the calibration and also validation tests in comparison to known standards. Section 3 presents the data for each set of experiments and Section 4 and Section 5 describe the implications of this work in the context of developing an actual instrument along with further improvements that can be made.

## 2. Methods

### 2.1. Machine Vision Position Tracking

Our machine vision approach utilizes a web camera mounted to one probe handle and a pair of AprilTags [29,30] printed on a flat surface of the second handle (Figure 1a). AprilTags are fiducial markers consisting of black and white block patterns (Figure 2) that are well established, robust, and popular for robotics and augmented reality applications [31]. The block pattern encodes a tag identification number (ID) and uniquely determines the identity of the four tag corners.

Since pose estimation is a common task in robotics, we configured our position tracking algorithm utilizing the Robot Operating System (ROS) framework [32]. Camera images arriving on the USB link are published to the ROS messaging middleware by an ROS camera driver node. Downstream, an ROS AprilTag detector node uses the AprilTag library (version 3.3.0) to detect and extract the ID and corner pixel locations of all visible tags from the images. This data is subsequently used by a TagSLAM (simultaneous localization and mapping) node [31] to compute the geometric transformation between the camera and its opposing tags. Finally, another ROS node leverages this transform and the calibrated geometry to compute the tip separation distance.

The optical pose estimation task presented here requires two forms of calibration—intrinsic (camera parameters) and extrinsic (geometry). For the intrinsic calibration we use the well-established open-source software package Kalibr [33] to determine the camera’s focal length, image center and lens distortion parameters.

With the intrinsic calibration parameters at hand and known tag size, the TagSLAM node can infer the pose of any tag with respect to the camera. It does so by means of a pose graph optimizer based on GTSAM (Georgia Tech Smoothing and Mapping) [34]. The correct tag pose is assumed to be the one for which the projected tag corner pixel coordinates agree optimally with the ones extracted by the AprilTag library. Note that we leverage optical pose estimation, not just to determine the geometric transformation between the handles during deployment, but also to facilitate extrinsic calibration as described below.

The extrinsic calibration requires (a) the position of the tip with respect to the camera coordinate system for the handle with the camera and (b) the location of the tip with respect to the tag coordinate system for the handle with the tags. Direct mechanical measurement of the tip positions is error prone and difficult, particularly for case (a) where the lens center and orientation cannot be directly measured. We overcome these issues by using a combination of optical and mechanical measurements. First, a calibration fixture is fabricated, as shown in Figure 3, consisting of a 3D printed post (label 1, green) glued to a surface on which an array of tags is also printed (label 2). Using mechanical measurement tools, we determine the position of the vertex at the tip of the calibration post with respect to the printed tag array. This is straightforward since the geometry is simple and all reference points are easily accessible.

Next, the handle with the camera is attached to the lab stand (see Figure 3, label 3) such that the camera faces the tag array, and the tip is precisely touching the vertex of the calibration post (not shown in Figure 3). TagSLAM is then used to determine the transform between the tag array and the camera, which, since the location of the tip is known with respect to the tag coordinate system, in turn permits calculation of the tip location in the camera reference frame (case a) above. We use a similar procedure to establish the tip position for (case b). Again, the tip of the probe is inserted into the calibration post vertex, but this time camera images are taken with both the tags on the probe handle and the tags of the tag array in view. This establishes the missing transformation between tag array and probe tags, allowing for the probe tip location recovery with respect to the handle tags.

It should be noted that the geometry of the tip itself is a small source of error in the measurements. Figure 4 shows a photographic close-up of the two tips close to each other. Our experience is that when the coaxes are inserted into the vertebra, they will be separated by an angle between 50 and 70 degrees. We chose to machine off the face of each tip at 45° each for mainly practical reasons. Angles much sharper than that would have increased the risk of the center conductor shorting to the outer conductor in the machining process. It would have also elongated the exposed portion of the center conductor and made it more difficult to compute the phase centers of each probe. Therefore, 45° was a practical compromise. Notwithstanding, the sharp point of the tip is not a perfect point but has a bit of a curve along one axis. When inserting the point into the calibration post vertex, there is some ambiguity as to what physical point the system is being calibrated to. The error associated with this uncertainty is included in the overall assessment in the Discussion section.

For realistic comparison purposes, the surgical navigation probes are built directly into the same probe handles (Figure 1). Each has a different sphere pattern so that they can be accurately distinguished from each other. The Polaris near infrared camera system was positioned close by so that the spheres could be identified by line of sight (Figure 5). We used a software package supplied by Polaris to calibrate the probes and compute their physical positions. The description of the associated process is not included here.

### 2.2. Design and Operation of Microwave Probes

As was described in Meaney et al. [21], the transmission-based probes operate by transmitting a signal from one open-ended coaxial aperture to another over a short distance. The microwave portion of the probes are fabricated directly from semi-rigid RG-405 coaxial cables. The cables have a 0.51 mm diameter beryllium copper center conductor, a 1.68 mm Teflon insulator and an overall outer diameter of 2.20 mm—the outer conductor is copper. Because the apertures are effectively in the far field of each other over our entire operating frequency range, the calculations of the phase and attenuation coefficients from the measurement data and subsequently the dielectric properties are relatively straightforward. In essence, the two factors which impact the microwave measurements are the properties of the medium and the separation distance. Calculation of the separation distance has been implemented using a simple machine vision concept, with the miniature camera incorporated into one of the probe handles while the encoded tags with the associated geometrical information are printed on the second handle. This is described in detail above. For calculation of the phase and attenuation coefficients, we need to calculate the phase and amplitude slopes as a function of propagation distance. This necessitates measuring the phase and magnitude for at least two different spacings. For the eventual in vivo operation, one measurement will be a reference outside of the body in a controlled setting—preferably with the probes nearly touching. The second will be with the probes inserted into the pedicle arms of the vertebra—utilizing the machine vision apparatus to derive the separation distance.

For the phase coefficient, the phase varies linearly in distance away from a radiator when in the far field. The major challenge is that the phase needs to be unwrapped for each measurement. As long as the phase is referenced to zero at the coaxial apertures, the phases for adjacent measurement frequencies are always within the range of −180° and +180° of each other. Exploiting this, it is easy to unwrap the phases for all frequencies. Moving the reference planes for each port of the VNA to the coaxial apertures is accomplished using the standard port extension feature found on all VNAs. The process involves holding the probe attached to Port 1 of the VNA in the air, monitoring the S11 phase for Port 1 and simply adding in an artificial delay line. One simply keeps adding in delay until the phase is as close to zero as possible for the entire frequency band. This is repeated for the second probe (attached to Port 2 of the VNA) and monitoring the S22 phase while adding in delay line until it is zero across the full frequency band. In basic circuit theory, the open-ended coaxial line is very close to a perfect open circuit (i.e., an S11 phase of zero) [35]. Once these two tasks are complete, calculating the phase coefficient is straightforward.

Calculating the attenuation coefficient is more complex because the signal strength decreases both linearly with respect to the effective plane wave attenuation in the medium but also as a function of 1/R^2^. The 1/R^2^ term needs to be subtracted out to reveal the actual attenuation coefficient. The challenge is to find the R = 0 location. For these probes, we have made a single important alteration compared with our original implementation where the perpendicularly oriented aperture faces (with respect to the axial orientation of the coax) were diametrically opposed to each other. However, this is not especially convenient for vertebrae application. In this case, we have opted for machining the face of the aperture on a 45° angle so that when the two are inserted into the pedicle holes, they are nominally facing each other. Even so, the probe apertures do not perfectly face each other, but this angle is a useful accommodation. It is important to note that the beam pattern is especially broad for these apertures given that their size effectively approaches a point source; therefore, the orientation of the faces can tolerate some deviation from a direct opposition registration without degrading the signal. The first approximation is the average to the distances between the oblique vertices on the probes with those for the sharper points at the ends. In practice this was a good initial approximation, but when comparing the attenuation coefficient with values derived from the known dielectric properties, there was usually a consistent offset. In this case, adding 1 mm to the values for R provided a more accurate agreement.

### 2.3. Calculation of Phase and Attenuation Coefficients from Dielectric Properties

As demonstrated in Meaney et al. [21], the dielectric properties can be derived from the phase and attenuation coefficients using the following relationships:(1)$$\varepsilon_{r}=\frac{\beta^{2}-{\;\alpha }^{2}}{\omega^{2}\mu_{o}\varepsilon_{o}}$$
(2)$$\sigma =\frac{2\beta \alpha }{{\omega \mu }_{o}}$$
where ε_r_, σ, ω, μ_o_ and ε_o_ are the relative permittivity, conductivity, radian frequency, free space magnetic permeability and the free space permittivity, respectively. Except for ω, these are the constitutive properties for Maxwell’s equations [36]. β and α are the phase and attenuation coefficients, respectively. In this case, both β and α are computed via slope relationships utilizing the measurement data. As can be seen from Equation (1), any variabilities in these are significantly magnified by the presence of the squared terms and the relative error further magnified from subtracting the two terms. Consequently, we have found that examining the data in terms of these coefficients is superior in terms of reducing ambiguity than opting for comparing the actual dielectric properties. For the purposes of comparing our measurements to ground truth, it is subsequently important to be able to compute the phase and attenuation coefficients from the actual dielectric properties. Equations (1) and (2) can be rearranged exploiting some algebra and the quadratic formula to yield:(3)$$\alpha =\sqrt{\left(\frac{\omega^{2}\mu_{o}\varepsilon_{o}\varepsilon_{r}}{2}\right)\left[-1+\sqrt{1+{\left(\frac{\varepsilon ”}{\varepsilon_{r}}\right)}^{2}}\right]}$$
(4)$$\beta =\left(\sigma \sqrt{\frac{\mu_{o}}{2\varepsilon_{o}\varepsilon_{r}}}\right)\left(\frac{1}{\sqrt{-1+\sqrt{1+{\left(\frac{\varepsilon ”}{\varepsilon_{r}}\right)}^{2}}}}\right)$$
where $\varepsilon ”\;=\frac{\sigma }{{\omega \varepsilon }_{o}}$.

### 2.4. 1 mm Thick Reference Material

For calculating the slopes of the magnitude and phase as a function of distance, we require two measurements at two different distances. The first is performed when the probes are inserted into the bone. There are several options for the second. One would be to have the second distance with the probe tips simply touching. This presents two problems: first, the coax inner and outer conductors could short out, which would substantially alter the measurements, and the second is that with the probes touching they would no longer be in the far field of each other. The ideal solution would be something with roughly a 1 mm thickness between the tips and reasonably mimicking the properties of the interrogated bone. We experimented with several different materials with 1 mm thickness but found the differences in phase and amplitude to be quite close—primarily because the signal propagates such a short distance between probes. Ultimately, we found that a simple, 1 mm thick piece of felt was convenient and provided similar results as the other materials.

## 3. Results

### 3.1. Dielectric Probe Measurements

Figure 6 shows a photograph of the two probe handles mounted to a fixture for performing the microwave measurements. Each was screwed onto a bracket constructed from a piece of aluminum angle iron which was then mounted to a plate with vertical and horizontal grooves. The dimensions of the grooves were chosen such that the coaxial cables of each handle were oriented roughly 25° from vertical—the overall angle between the two coaxes was twice that or 50°. The grooves on the left were oriented vertically to allow for accurate alignment between the two coaxial tips. The grooves to the right were designed to allow for adjusting the separation distance between the two probes.

The probe connectors were connected to the ports of the Rohde and Schwarz ZNBT8 vector network analyzer (VNA) (Munich, Germany) (behind the setup) via double braided coaxial cables. Data was acquired over the full frequency range from 100 kHz to 8.5 GHz using 201 frequencies. The VNA was calibrated utilizing a standard calibration kit for the reference plane associated with the ends of the two flexible coaxial cables. The IF bandwidth was set to 10 Hz to maximize the dynamic range. We subsequently utilized the port extension feature to translate the reference plane to the ends of the two probes. In this case, the impedance at the probe tips was nominally an open circuit (when operating in air). We subsequently added delay lengths until the S11 and S22 measurements closely resembled an open circuit—in this case with the phase nominally zero across the full frequency range (Figure 7). There is some ripple in the phase plots which increases progressively for higher frequencies, but the perturbations are relatively minor.

We acquired data for the probes submerged in different concentrations of glycerin and water: 20%, 50% and 80% glycerin concentrations. The lengths of coax were sufficiently submerged to minimize the multi-path signals except for the low end of the spectrum. To mimic the conditions we anticipate when operating in vivo, we took amplitude and phase measurements for when the probe tips were nearly touching, except for a thin layer of felt between them. The felt was held in place by an assistant. We subsequently acquired data for five different probe separation distances that were representative of what we expect in the clinic—1.6, 1.8, 2.0, 2.2 and 2.4 cm. Figure 8 shows plots of the raw magnitude and phase measurement data for the felt case and the other five distances for the case with the probes submerged in the 20% glycerin solution. As can be seen, there is a monotonic decrease in both the magnitude and phase for the majority of the frequency band. It should be noted that the data below roughly 2 GHz is corrupted by multi-path signals, while that for higher frequencies is substantially affected by the noise as the signal levels decrease. For display purposes, the phase data for the instances where the signal goes below the noise floor are not shown and ultimately not used in the subsequent analysis.

Figure 9 shows the slopes of the magnitudes (attenuation coefficient) and phases (phase coefficient) with respect to separation distance as a function of frequency for the five separation cases in the 20% glycerin bath. In this case we display the data only over 2–5 GHz to filter out the aberrant characteristics where the signals are corrupted by either multi-path signals or noise. We also overlay the exact values derived from the exact property values measured with a Keysight Dielectric Probe Kit (85070E—Santa Rosa, CA, USA). For each of the five cases, the slopes were computed using measurements at the separation distance and with the felt separation, as would be the case in vivo. It should be noted that the 1/R^2^ factor has been removed from the magnitude measurements before calculating the slopes. Nominally R should be calculated as the distance between the phase centers of each probe. To estimate this, we physically measured the distance between the closest portion of the tips and the furthest points (i.e., the sharp tip) and took an average of the two. We noted that there was a slight offset between the actual and measured magnitude slopes which was attributed to the R distance being slightly off. For these cases, we adjusted R by 1 mm. We also used a mean filter to smooth out the small ripples in the data.

The phase data presented has been unwrapped as a function of frequency. In most of our previous work, we were able to assume that the phases at the lowest frequencies resided in the Riemann sheet between −180° and 180°. However, because of the multi-path corruption, the lowest-frequency data was inconsistent and unreliable. In this situation, we started with the values at 2 GHz and worked upwards and downwards. It should be noted that for the different separation distances, it was not always assured that the 2 GHz phases would reside between −180° and 180°. It is crucial to be able to determine this in a robust and reliable manner. We have developed a simple procedure for this which is the subject of a separate manuscript.

The attenuation and phase coefficient plots are quite consistent for the five cases and match that of the actual values quite well—both in terms of the absolute values but also with respect to the associated dispersion characteristics. Figure 10 shows the same coefficient plots except for all three liquids along with their associated exact values. As can be seen, the phase values easily differentiate the different liquids. The attenuation coefficient plots are more complex because of the varying dispersion characteristics. In fact, for the 50% and 80% glycerin concentration cases, there is a crossover at roughly 3 GHz. Even with this level of complexity, the values derived from the transmission probes track the actual values quite well.

### 3.2. Separation Distance Measurements and Comparison

Figure 11 shows a scatter plot of the differences between the probe-based separation distances and the mechanically measured separation distances as a function of the mechanically measured distances for the machine vision concept. The separation distances ranged from 5 to 120 mm, and the angles between the probes ranged from roughly 30° to 90°. In this case there is a small but consistent bias, i.e., our optical method reports a separation distance that is slightly larger than mechanically measured. We discuss possible sources of this bias in Section 4. However, after subtracting the mean error, the remaining standard deviation is only 0.4 mm over a wide range of separation distances. A similar comparison was performed for the surgical navigation system and the same separation distance plots are shown in Figure 12. The error in this case was 1.6 mm.

## 4. Discussion

This paper demonstrates that the two key probe components, the integrated machine vision apparatus and transmission dielectric probes, work well and are suitable for use as a vertebrae strength probe. The machine vision concept is based on a TagSLAM configuration that can track the position of the probe on the tag handle and subsequently compute the separation distance between the two probes. A suitable calibration routine was developed exploiting a commercially available software package which configured both the intrinsic and extrinsic features. Tests were performed to assess the accuracy for a pair of single separation distances but with different probe angle orientations to assess the algorithm robustness. In both cases, the recovered standard deviations of the distances were less than 0.4 mm. This compares favorably to the results using the surgical navigation system. From a practitioner standpoint, the errors for tracking a single point are generally below 0.5 mm. We most likely encountered larger values because of the length and flexibility of the coaxial portion of the probe. However, a more significant aspect relates to the fact that the surgical navigation system has to track two separate probe tips and then compute the separation distance via a subtraction. In this situation, the error for tracking the tip locations is doubled when computing the distance. This is directly reflected in the fact that the standard deviation of the distances using the Polaris system was on the order of 1.6 mm.

There is some ambiguity of the computed probe tip determination, which is most likely due to the fact that the tip is not an exact point but rather a slightly rounded edge of the cylindrically shaped coaxial cable cut at an acute angle. The manner in which it fits into the vertex of the three-post orthogonal axis fixture adds to this ambiguity. The standard deviations recorded for a range of angular orientations are encouraging. In practice, we expect the angular range to be considerably narrower given the consistent geometry of the vertebrae and a narrower and predictable variation.

Regardless of which tracking concept is used, both will benefit from two simple improvements in the probe design. The first is that the coaxial portions of the probes do not need to be as long as that used for this experiment. Even small stresses from touching the tip on any object can cause minor position deflections that can add to the error. To further assist in this, we propose to add a cone-like stiffener that would slide directly over the coax with the larger diameter closer to the handle and tapering to the coax diameter nearer the tip. It will need to terminate prior to the probe tips since there is only space for the coaxes to fit inside the pedicle guide holes.

One important concern will be whether the devices will be made to be reusable or disposable. Clearly the measurement VNA will need to be reused. In fact, it will need to remain outside of the sterile field during surgery. The more complicated aspect will be to determine the configuration of the handles. It is entirely possible for these probes to be disposable. The semi-rigid coaxial cable and pair of connectors are quite inexpensive. Likewise, the current handles are 3D printed and could be fabricated in a lower-cost manner in larger quantity. The most complicated feature is the camera and its associated circuit board. These are relatively inexpensive—less than $20 each. At a minimum they will need to be sterilized at least once prior to use. Repeated sterilization could prove difficult and may point to the disposable option. This will be evaluated as part of the future development.

Compatibility of the probe with human tissue is a major concern. Interestingly, this could actually be a limiting factor for the reflection-based probe. The most obvious way to make the coaxes tissue compatible is to cover them with some type of polymer such as parylene [37]. However, for the calibration process for the reflection probes, it is essential to use a short-circuit as one of the calibration standards. This would be impossible because the ends of the coax would be completely covered with the polymer. Conversely, the impact of the parylene would have an almost imperceptible impact of the transmission-based measurements. As mentioned earlier, the interface spaces between the tissue and the associated pair of probes comprises only a small fraction of the tissue being interrogated, i.e., that residing in the space between the probe tips. Even small discontinuities at these interfaces have minimal impact because they comprise a small fraction of the physical space between the probes. Hence, from an electromagnetic perspective, a parylene coating will have minimal effect. The more crucial aspect will be in determining whether parylene or another coating will be suitable and sufficiently reliable for human use. This will be a critical concern in the next stage of development.

In addition to compatibility, electrical safety is important. The VNA will only transmit 1 mW of RF power, which is well below the safety standards for this frequency range [38]. It is also critical to eliminate the possibility of electrical shocks. There will be a DC block on the output of the VNA to restrict any shock through the coax. We will also design the power supply for the camera such that it doesn’t pose a threat to patient safety.

Finally, one of the key challenges of the final design will be the need to take into account the workflow aspects of the surgical procedure. The technology will ultimately include the two probe handles (with integrated camera), the VNA and a computer with video output for collecting and processing the data. The VNA will need to remain outside of the sterile field, as would a stand-alone computer. One consideration would be to utilize a compact Raspberry PI computer that could be integrated directly into the handles. It could also eliminate the need for a USB connection to the camera. Its additional cost would be modest, so it would still be possible to fabricate the device as disposable. The design will also need to take into consideration what calibration processes can be performed a priori (possibly even at the manufacturing stage) and what needs to be done at the time of surgery. These discussions will need to be made early in the design process and involve the surgeons to help optimize the overall workflow.

## 5. Conclusions and Future Work

We are developing a transmission-based microwave probe for assessing bone strength in vertebrae as part of the spinal fusion surgical procedure. We have validated the feasibility of the two critical features of the device: (a) the phase and attenuation coefficients derived from using the probes produce similar values as those for conventional reflection-based probes when tested in known dielectric liquids, and (b) the machine vision concept for calculating the separation distances of the two probe tips is accurate to within 0.4 mm and is superior to a surgical navigation technique. These innovations will make it possible to finalize the design of a low-cost, compact and convenient probe that can be used in spinal fusion surgeries.

It is important that we benchmark the performance of the machine vision concept with respect to that for the surgical navigation technique. The next stage of development for this project will be to optimize the design so that it is compatible for actual human use. Within the realm of using this in an operating room, several key aspects need to be considered: (1) is the accuracy sufficient, (2) is the configuration sterilizable, and (3) can the process be integrated into the existing workflow without added time expense. For the machine vision method, we have sufficiently demonstrated that its accuracy is superior to that of the latter and that the measurements should be able to be performed quickly. However, the introduction of the digital camera does raise questions regarding sterilizability. For the surgical navigation approach, it should also be able to be performed quickly. It does have the added advantage that surgical navigation techniques are now widely accepted in operating room settings—both with respect to sterilization and workflow accommodation. The challenge will be to assess whether opting for the machine vision technique with its superior performance is worth the extra development problems needed to be overcome with respect to sterilization.

A key question will be to assess what level of accuracy is sufficient for adequate delineation of normal and osteoporotic bone. While there is some published data relating to the dielectric property dependence on bone mineral density, this work was performed on excised animal bones [15,39]. We are currently experimenting with excised hip samples for normal and osteoporotic patients. Future tests are planned to test the optimized design on actual vertebrae during surgery. This will require design refinements to accommodate the sterilization along with the necessary real-time calibration of the technique in the OR. We are currently exploring funding for this follow-on effort.

Beyond the vertebrae testing applications, exploiting tissue dielectric properties can have substantial impact on medical testing. As alluded to in the Introduction, while benchtop dielectric probe techniques have been available for decades, they have not advanced into surgical applications in spite of the significant specificity that the dielectric properties offer. These probes overcome the logistical challenges of the commercial reflection-based probes while still exploiting the dielectric property-based differentiation. For instance, there is the opportunity to assess hip bone strength as a means of differentiating between bone requiring full replacement and just augmentation. In addition, we are also exploring using the transmission technique with the coaxes positioned side-by-side instead of across from each other [40]. In this case we have modified the coaxes via metal 3D printing technology to help assist in increasing the probe penetration depth. This technology is currently in a pilot clinical trial for diagnosing and detecting sarcopenia.

## Figures and Tables

**Figure 1 sensors-23-04819-f001:**
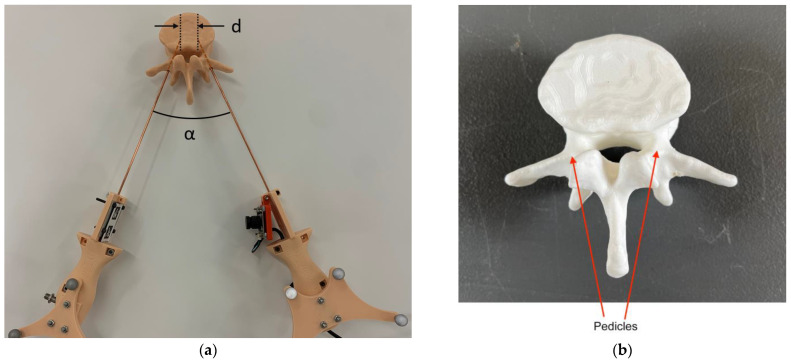
(**a**) Photograph of the probe handle pair with the associated camera and AprilTags on opposing handles. The coaxial tips are inserted through holes in the pedicles and subsequently into the main body of the vertebra. α represents the angle between the two coaxes and d represents the spacing between the probe tips. (**b**) Close-up of the vertebra with arrows indicating the location of the pedicles.

**Figure 2 sensors-23-04819-f002:**
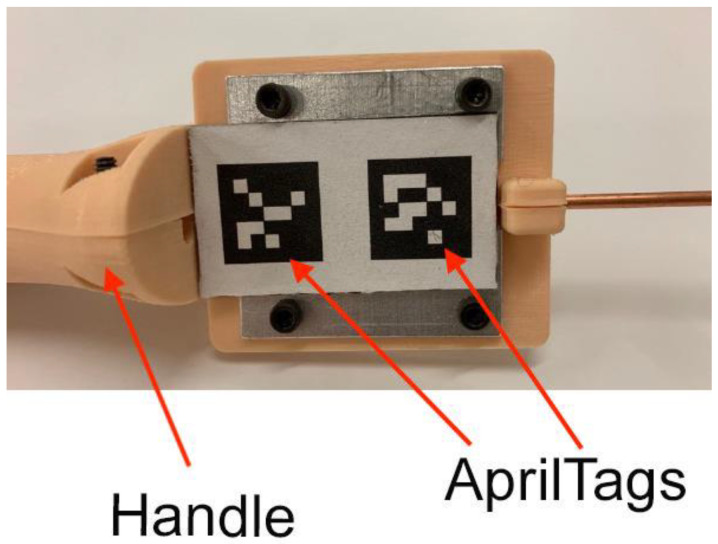
Photograph of a pair of printed AprilTags mounted to a flat surface on one of the probe handles. Arrows are used to point to the AprilTags and the probe handle.

**Figure 3 sensors-23-04819-f003:**
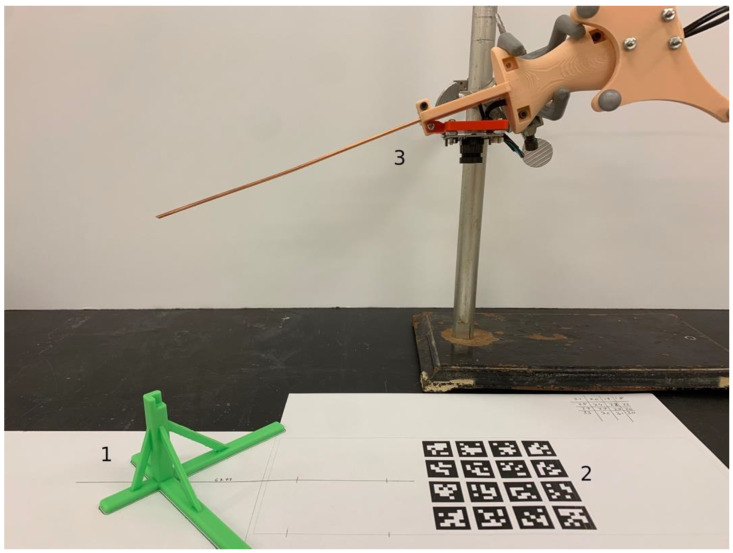
Photograph of the tag array with respect to the camera on the probe handle. The 3D printed green post is the exact mechanical reference for extrinsic calibration. “1” indicates the 3D printed post, “2” indicates the array of reference tags, and “3” indicates the handle with the camera attached to the lab stand.

**Figure 4 sensors-23-04819-f004:**
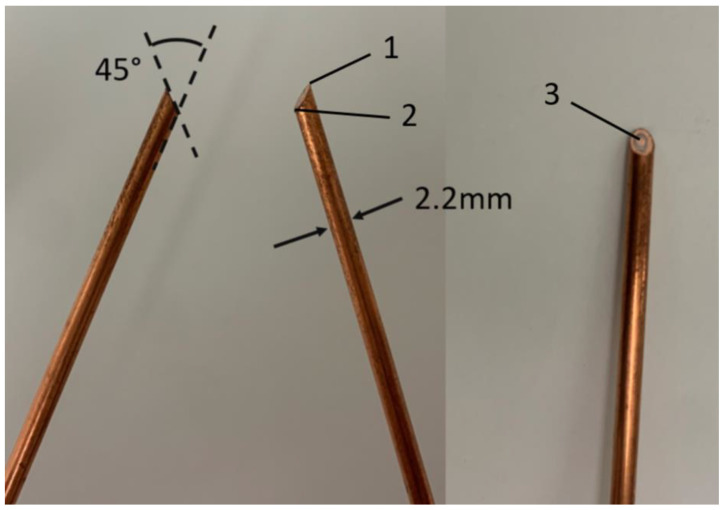
Photographs of the probe tips from different views. In this case, the probe tips are machined at a 45° angle from the axis of the coax. Points 1 and 2 are the associated acute and obtuse angles, while Point 3 shows the slightly elongated center conductor after machining.

**Figure 5 sensors-23-04819-f005:**
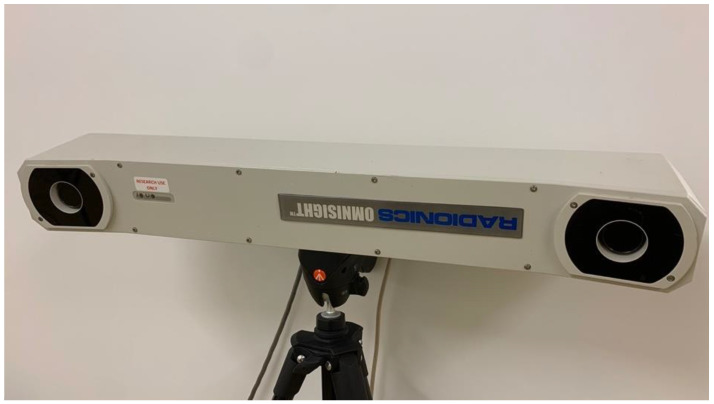
Photograph of the Polaris near infrared camera system for tracking objects with optically coated spheres.

**Figure 6 sensors-23-04819-f006:**
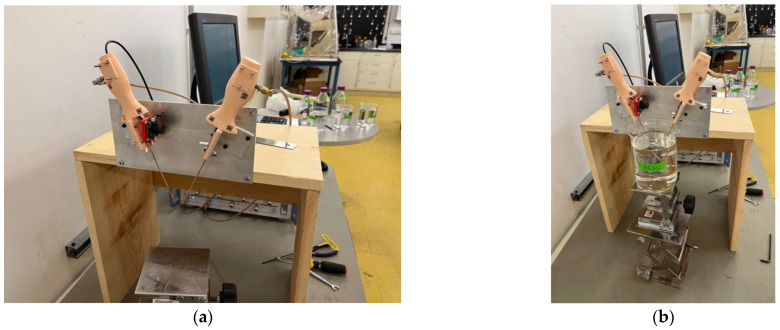
(**a**) Photograph of the two probe handles mounted to the experimental fixture. The coaxial probes are each oriented at 25° to vertical, and the web camera for the left handle is aimed at the tags (not shown) printed on a face of the right handle. The aluminum fixture allows for the handles to be moved vertically (left) or horizontally (right) while maintaining the same angle orientation. (**b**) Photograph of the probes with the coaxes submerged in a glycerin–water mixture.

**Figure 7 sensors-23-04819-f007:**
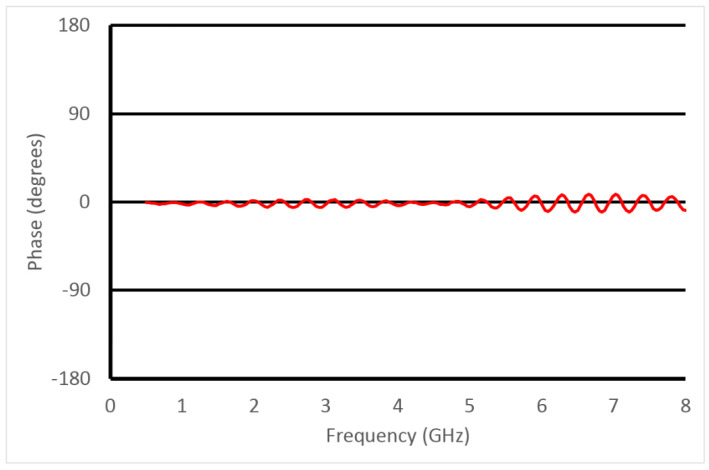
Plot of the S11 phase as a function of frequency for probe 1 after the port extension procedure. Zero phase for S11 is nominally an open circuit, which the probe tip closely resembles.

**Figure 8 sensors-23-04819-f008:**
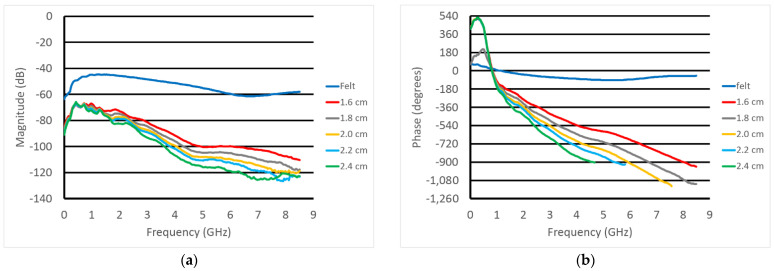
Plots of the (**a**) magnitude and (**b**) phase as a function of frequency for the reference measurement and five different probe tip spacings. Note that the phases have been unwrapped.

**Figure 9 sensors-23-04819-f009:**
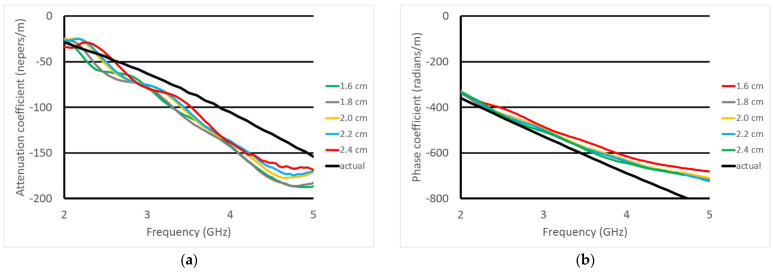
(**a**) attenuation and (**b**) phase coefficients as a function of frequency for different probe tip separation distances. The values based on the actual liquid dielectric properties are also plotted for reference.

**Figure 10 sensors-23-04819-f010:**
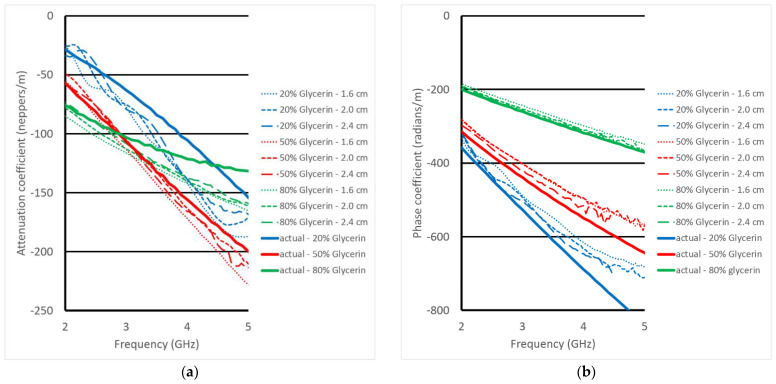
(**a**) attenuation and (**b**) phase coefficients as a function of frequency for different probe tip separation distances and three different dielectric liquids—20%, 50% and 80% glycerin mixtures. The values based on the actual liquid dielectric properties are also plotted for reference. Note that the plots for only three distances (1.6, 2.0 and 2.4 mm) are shown for each concentration to improve clarity.

**Figure 11 sensors-23-04819-f011:**
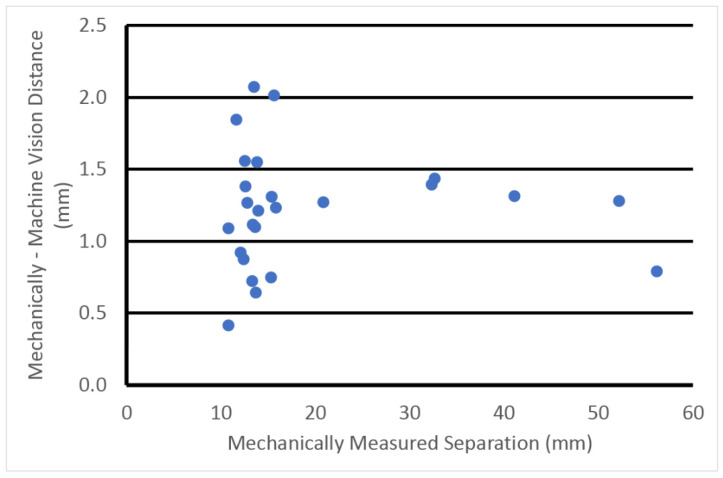
Scatter plot of the differences between the mechanically measured and machine-vision-measured separation distances with respect to the mechanically measured separation distances.

**Figure 12 sensors-23-04819-f012:**
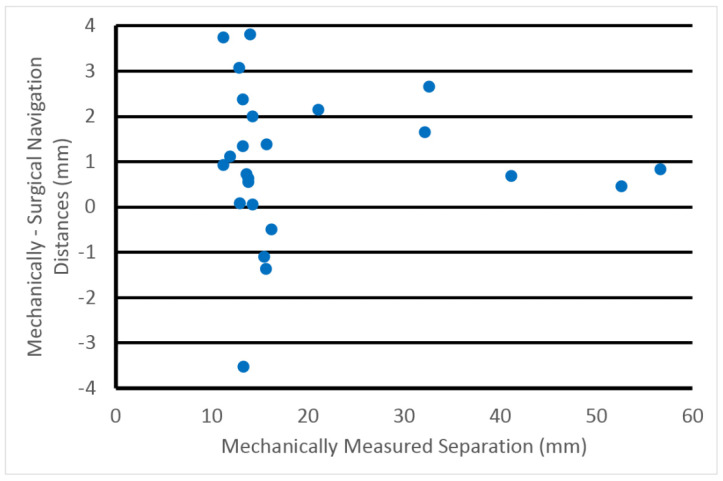
Scatter plot of the differences between the mechanically measured and surgical-navigation-measured separation distances with respect to the mechanically measured separation distances.

## Data Availability

Data is available upon request.

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
