# Peer review of "Transmission-Based Vertebrae Strength Probe Development: Far Field Probe Property Extraction and Integrated Machine Vision Distance Validation Experiments"

_sensors, 2023, doi:10.3390/s23104819_

Round 1

Reviewer 1 Report

The content of the article is consistent with the scientific area of the journal Sensors. The subject raised by the authors is current and so far rarely noticed by other authors publishing in this area.
The issue described may in the future contribute to improving the efficiency of the automation of the instrumentation or machine vision and surgical navigation.
The paper is of an original scientific nature, related to the development of a vertebral strength probe based on transmission technology, i.e. machine vision distance verification. The devise is based on a transmission probe whereby thin coaxial probes are inserted into the small canals through the pedicles and into the vertebrae and a broad band signal is transmitted from one probe to the other across the bone tissue.
For a better clarification, please edit your paper as follows:
1. Expand the text of the manuscript (or the introduction or conclusion) with specific results in the world and in Europe, - increase the quality of the work by listing the results of publications of researchers and experts working in this field registered in world databases - wos. These are: Visual Product Inspection Based on Deep Learning Methods, Collaborative assembly task realization using selected type of a human-robot interaction, thanks.
2. figures 1 and 6a should be contrasting and readable,
3. conclusions and future work should be extended to contain practical applications based on research described in this paper - expand references,
4. highlight the course of dependencies/relations in figure No. 11 - the yellow color is indistinct,
5. For ARTICLE type, 24 references are not enough. Please add more
references (>24) during your revisions.
I recommend publishing the post after the proposed modifications.

Author Response

Dear reviewer,

Thank you for your thoughtful and constructive criticisms.  Below we answer the questions raised and corresponding changes have been made to the revised text.

The content of the article is consistent with the scientific area of the journal Sensors.  The subject raised by the authors is current and so far rarely noticed by other authors publishing in this area.

The issue described may in the future contribute to improving the efficiency of the automation of the instrumentation or machine vision and surgical navigation.  The paper is of an original scientific nature, related to the development of a vertebral strength probe based on transmission technology, i.e. machine vision distance verification. The devise is based on a transmission probe whereby thin coaxial probes are inserted into the small canals through the pedicles and into the vertebrae and abroad band signal is transmitted from one probe to the other across the bone tissue.  For a better clarification, please edit your paper as follows:

  1. Expand the text of the manuscript (or the introduction or conclusion) with specific results in the world and in Europe, - increase the quality of the work by listing the results of publications of researchers and experts working in this field registered inworld databases - wos. These are: Visual Product Inspection Based on Deep Learning Methods, Collaborative assembly task realization using selected type of a human-robot interaction, thanks.

We have added references to these topics in the Introduction.

  1. figures 1 and 6a should be contrasting and readable,

We have eliminated Figure 6a (now 5a) because it was essentially the same as Figure 1.  The figure references in Section 2.1 now refer to Figure 1a.

In this case, Figure 1a shows a photograph of the probes, their handles and the associated camera and April tag.  The key feature of this is to demonstrate the orientation of the probes to the holes in the vertebrae along with the distance inside of the bone separating the two probe tips.  This photo is clear and the added dimensioning appropriate.

  1. conclusions and future work should be extended to contain practical applications based on research described in this paper - expand references,

We have expanded the Conclusions section into the Conclusions and Future Work section.  The first new paragraphs describe the steps that will be needed to turn the prototype transmission probes into an actual clinical device.  Beyond that, we discuss possible future opportunities.  For instance, alternative ways of measuring bone could be useful such as the hip where surgeons may opt to simply augment the current bone versus replace it.  These applications could involve somewhat invasive processes like drilling holes for inserting the probes into.  We also discuss the opportunities for exploiting the transmission concept except putting the two coaxial cables side-by-side instead of across from each other.  Work has already started in this area for sarcopenia diagnosis and assessment which as reach the stage of pilot clinical trial.

  1. highlight the course of dependencies/relations in figure No. 11 - the yellow color is indistinct,

We’ve changed the figures considerably for clarity.  First, we have only included the plots for three separation distance for each concentration to improve clarity.  Second, we have grouped the sets of plots of different concentrations by color – blue, red, and green.  Third, for each concentration, the actual value plot is a large solid line while the computed values for the different separations are a thinner line and each is distinguished by a different dash type.  This allows the reader to more qualitatively assess how well each group of plots for a given concentration tracks the associated actual value plot.  For one plot, phase coefficient for the 80% glycerin concentration, the curves are relatively closely packed.  It is challenging to see each individual curve for this.  However, this only confirms how close the computed values are to the actual.  In sacrificing a little clarity here, it allows us to make it clearer for the other concentrations.

  1. For ARTICLE type, 24 references are not enough. Please add more references (>24) during your revisions.

We have added 16 new references that help clarify key points.

I recommend publishing the post after the proposed modifications.

Reviewer 2 Report

Manuscript entitled “Transmission-based vertebrae strength probe development: probe property extraction and machine vision distance validation experiments” gives an interesting and systematic research about development of a transmission-based probe device for point-of-care assessment of vertebrae strength needed for fabricating the instrumentation used in supporting the spinal column during spinal fusion surgery. Additionally, machine vision scheme has been developed to measure the separation distance between the probe tips while they are inserted into the vertebrae.

The topic is original and relevant in the field, because the number of spinal fusion surgery has exploded in recent years.

The manuscript is well organized, the authors used the scientific methods, and they are adequately described. The results are clearly described and the discussion part is well presented. At the end of the manuscript, authors declare that those results, presented in this manuscript, will enable to finalize the design of a low cost, compact and convenient probe that can be used in spinal fusion surgeries.

 The references cited in the manuscript are recent, mostly within the last ten years. The most of the figures and images are presented appropriately and clearly. Data presented in charts are properly presented and easy to interpret and understand. The most of the concluded observations are written in the discussions that I find appropriate for this research.

 The acceptance of the manuscript will be suggested after the further corrections will be made:

- In the text, there is an unneeded space in several places that has to be removed.

- Image 3 is blurry. It should be replaced with a sharp, clearly visible image.

- It is necessary to describe the creation of AprilTags in more detail. In particular, how was the selection of tags determined, in what way and on what substrate were the tags printed. The authors claim that the presented model is ideal (in the sentence: “It uses an array of AprilTags (Figure 3), the full array size, individual tag border size and the camera model to produce an ideal mapping structure”), so it is necessary to describe the basis on which they came to this findings.

Author Response

Dear reviewer,

Thank you for your thoughtful and constructive criticisms.  Below we answer the questions raised and corresponding changes have been made to the revised text.

Manuscript entitled “Transmission-based vertebrae strength probe development: probe property extraction and machine vision distance validation experiments” gives an interesting and systematic research about development of a transmission-based probe device for point-of-care assessment of vertebrae strength needed for fabricating the instrumentation used in supporting the spinal column during spinal fusion surgery. Additionally, machine vision scheme has been developed to measure the separation distance between the probe tips while they are inserted into the vertebrae.

The topic is original and relevant in the field, because the number of spinal fusion surgery has exploded in recent years.

The manuscript is well organized, the authors used the scientific methods, and they are adequately described. The results are clearly described and the discussion part is well presented. At the end of the manuscript, authors declare that those results, presented in this manuscript, will enable to finalize the design of a low cost, compact and convenient probe that can be used in spinal fusion surgeries.

The references cited in the manuscript are recent, mostly within the last ten years. The most of the figures and images are presented appropriately and clearly. Data presented in charts are properly presented and easy to interpret and understand. The most of the concluded observations are written in the discussions that I find appropriate for this research.

The acceptance of the manuscript will be suggested after the further corrections will be made:

- In the text, there is an unneeded space in several places that has to be removed.

We have removed all unnecessary spaces between the lines of text, captions, figures and equations.  The large white areas to the left of the main text are part of the template provided by the publisher.

- Image 3 is blurry. It should be replaced with a sharp, clearly visible image.

Image 3 has been deleted.  It was essentially redundant with the array shown in Figure 4 (now Figure 3).  This is part of a more substantial re-write of Section 2.1.

- It is necessary to describe the creation of AprilTags in more detail. In particular, how was the selection of tags determined, in what way and on what substrate were the tags printed. The authors claim that the presented model is ideal (in the sentence: “It uses an array of AprilTags (Figure 3), the full array size, individual tag border size and the camera model to produce an ideal mapping structure”), so it is necessary to describe the basis on which they came to this findings.

Section 2.1 has been re-written in its entirety.  This includes a thorough description of how the technique works with several new references related to it.  I should note that the tags were printed on a simple piece of paper and then glued to a 1 mm thick sheet of smooth fiberglass board.

Reviewer 3 Report

Reviewer’s comments:

Meaney et al. reported the application of a previously established transmission-based coaxial probe approach in measuring vertebra strength.  This study has significant value and potential in improving the clinical care.  The manuscript is reasonably well written and organized.  Therefore, I recommend acceptance with minor revision, with detailed comments listed below:

1.        Please rephrase the title of the paper to better summarize the study and also easier to follow.

2.       In Line 36, please spell out DXA before using the abbreviation in the subsequent text.

3.       In Line 49, the authors could change it to “for quick and reliable assessments of the vertebrae strength”

4.       In Line 68, there seems to be a different font style or size.  

5.       Fig. 1: please label what d and α stand for (distance and angle?).  Please indicate pedicles in the photo.

6.       Fig. 2: please use arrows and texts to indicate the AprilTags and probe handles in the photo.

7.       Fig. 4: please include in the legend what 1, 2, and 3 each is corresponding to.

8.       In Line 241 -242, please enlarge the font used for the formulas to make all the letters legible.

9.       Fig 8: please include illustrations of how the 0 reading of the S11 phase is obtained.

1.   Fig. 9:  in the graph labeling, what does “felt” stand for?

1.   Related to the data shown in Fig. 11, glycerin is an interesting medium to use.  I wonder if the authors tried using postmodern vertebrae to calibrate and demonstrate the accuracy of the probes.

1.   The authors need to discuss the safety concerns related to inserting the probes into pedicles at the depth needed.

Decent

Author Response

Dear reviewer,

Thank you for your thoughtful and constructive criticisms.  Below we answer the questions raised and corresponding changes have been made to the revised text.

Meaney et al. reported the application of a previously established transmission-based coaxial probe approach in measuring vertebra strength. This study has significant value and potential in improving the clinical care. The manuscript is reasonably well written and organized. Therefore, I recommend acceptance with minor revision, with detailed comments listed below:

  1. Please rephrase the title of the paper to better summarize the study and also easier to follow.

For this project, the two main points that the paper makes is that we can exploit the far field to derive the material microwave properties and that we have developed a machine vision concept that assists in recovering those properties.  We have slightly altered the title to emphasize that the properties come from these far field calculations.

  1. In Line 36, please spell out DXA before using the abbreviation in the subsequent text.

Done

  1. In Line 49, the authors could change it to “for quick and reliable assessments of the vertebrae strength

Done

  1. In Line 68, there seems to be a different font style or size.

We’ve checked this and it looks fine.

  1. Fig. 1: please label what d and α stand for (distance and angle?). Please indicate pedicles in the photo.

Text has been added to the caption to describe what d and α stand for.  In addition, a separate photo of the vertebrae has been added to more clearly show where the pedicles are.

  1. Fig. 2: please use arrows and texts to indicate the AprilTags and probe handles in the photo.

We’ve added arrows to indicate these.

  1. Fig. 4: please include in the legend what 1, 2, and 3 each is corresponding to.

Done

  1. In Line 241 -242, please enlarge the font used for the formulas to make all the letters legible.

Done

  1. Fig 8: please include illustrations of how the 0 reading of the S11 phase is obtained.

We have added some text to elaborate on this.  Basically we simply hold the probe in air with nothing close to the tip.  We then monitor the S11 (i.e. reflected) phase on the VNA and adjust the port extension length until the phase is very nearly zero.  We also repeat the process for the second probe.  This is a very standard function on all VNA’s that we are aware of.  We’ve added a Keysight application note as a reference.

  1. Fig. 9: in the graph labeling, what does “felt” stand for?

Thank you for bringing this up.  As indicated in the paper, for calculating the slopes of the magnitude and phase as a function of distance, we require two measurements at two different distances.  The first is when the probes are inserted into the bone.  One possible solution would be to have the second distance with the probe tips simply touching.  This presents two problems:  First, the coax inner and outer conductors could short out with each other which would substantially alter the measurements, and the second is that with the probes touching, they would no longer be in the far field of each other.  Ideally, we wanted something with about a 1 mm thickness between the tips.  We experimented with several different materials, but found the difference in phase and amplitude to be quite close – primarily because the signal propagates such a short distance between probes.  Ultimately, we found that a simple piece of felt was convenient and provided similar results as the other materials.  We plan explore ways to make this more systematic in our future developments.  We’ve added a short subsection (2.4) to clarify this.

  1. Related to the data shown in Fig. 11, glycerin is an interesting medium to use. I wonder if the authors tried using postmodern vertebrae to calibrate and demonstrate the accuracy of the probes.

This is, in fact, one of our next experiments.  We’ve made mention of this in the expanded Conclusions and Future Work section.

  1. The authors need to discuss the safety concerns related to inserting the probes into pedicles at the depth needed.

There are three primary concerns regarding the safety of these new probes: (a) compatibility of the probe materials, (b) microwave signal safety, and (c) the potential for electric shock.  We have already addressed the issue of material compatibility, primarily by considering the use of a parylene film to cover the probes as discussed in the Discussion section.  The microwave signals used will always be below 1 mW.  This is well within the safety standards and is roughly three orders of magnitude below the power levels emitted by a standard cell phone.  We have added a reference for this.  For the electrical safety, this will be a subject for further development.  There will be no path for DC electricity from the VNA.  The only viable source will be from the camera. 

Reviewer 4 Report

The paper presents a devise based on a transmission probe whereby thin coaxial probes are inserted into the small canals through the pedicles and the vertebrae. Then, a broadband signal is transmitted across the bone tissue from one probe to the other.

The manuscript requires some significant changes:

As a general observation, you should check the spacing between sentences (phrases) because there seems to be more than one space between them.

1. Introduction

- Why is this study of interest? What is the defining part of this study that makes it important for research?

- The contributions and importance of the paper should be highlighted. What are the main advantages of this article concerning other similar works?

- Figure 1 does not respect the correct formatting according to the MDPI format.

2. Material and methods

- Figures 2-5 do not respect the correct formatting according to the MDPI format.

- Equations should be processed so that notations and indices can be easily identified.

3. Results

- Figure 7 does not respect the correct formatting according to the MDPI format.

- Figures 9, 10, and 11 should be redone because they seem unclear, and the associated legends are hard to read.

- Figures 12 and 13 do not respect the correct formatting according to the MDPI format.

4. Discussion

In section 4, the results need to be discussed and considered in other studies. Therefore, please refer to these studies. Also, it would be best to describe your research's limitations.

5. Conclusion

Here, in addition to conclusions regarding the obtained results, you must also present the following:

- What are the future works?

- What are the main challenges of the current work?

Reference

References should be more extensive and up-to-date. At least 70% must be from the last 10 years.

Author Response

Dear reviewer,

Thank you for your thoughtful and constructive criticisms.  Below we answer the questions raised and corresponding changes have been made to the revised text.

The paper presents a devise based on a transmission probe whereby thin coaxial probes are inserted into the small canals through the pedicles and the vertebrae.  Then, a broadband signal is transmitted across the bone tissue from one probe to the other.

The manuscript requires some significant changes:

As a general observation, you should check the spacing between sentences(phrases) because there seems to be more than one space between them.

All double spacing has been eliminated.

  1. Introduction

- Why is this study of interest? What is the defining part of this study that makes it important for research?

There are several key points that demonstrate why this research/development is important.  First, there is an unmet need.  Right now the only way surgeons can screen for determining if the vertebrae are healthy enough to withstand the screw strain is DXA x-ray scans prior to surgery.  There are too many surgical failures related to the screws pulling out from the vertebrae – the reasons primarily relate to the presence of osteoporosis.  There are numerous problems with DXA in this context as is described and summarized in the text.  This is discussed in the first paragraph of the Introduction.

There is a need for alternative, and preferably point-of-care solutions for this.  The challenge is that the holes available to get a probe into are very small.  Optical techniques would not be viable because the signal attenuation through the bone would be too great.  We have added a sentence related to this.  There is a commercial, microwave reflection-based probe which could fit in the holes.  The dielectric properties of normal and osteoporotic bone are sufficiently different that a dielectric probe could be useful in this setting.  However, we have pointed out in the text that the poor penetration depth and the unwanted artifacts from moving the cables make this an untenable choice.  This has been described in the text.  We are not aware of potential alternatives.  This is discussed in the second paragraph of the Introduction.

This is an ideal opportunity for our transmission-based probe.  First, from a configuration standpoint, this is suitable because we can put separate probes into the two holes through the pedicles and into the vertebral body.  Second, we have previously shown that the penetration depth of this probe can easily deal with having to transmit across the 1-2.5 cm separation distance we anticipate in actual vertebrae.  We have also shown that we can derive the dielectric properties from this style of measurements.  This is discussed in the third paragraph of the Introduction.

After all of that preliminary background, the key investigation of this paper is whether this concept can be integrated into a simple pair of probes.  For this investigation, there are two key points that need to be demonstrated before progressing to the next stage of development.  First, is whether the microwave measurements from this simple configuration can be used to recover the dielectric properties.  Previous experiments were simpler with the open-ended coaxial probes directly facing each other.  The second is that we need to be able to accurately compute the distance between the probe tips when they are inserted into the vertebrae.  That is the whole reason for developing the machine vision-based and the surgical navigation-based approaches.  The experiments here quantified the accuracy which each approach could reliably achieve.  These are the two key building blocks necessary before progressing to testing on actual ex vivo human bone and eventually in human experiments.  We feel that these points have been presented in a logical sequence in the manuscript.

- The contributions and importance of the paper should be highlighted. What are the main advantages of this article concerning other similar works?

As mentioned above and summarized much more briefly here, there is a clinical need for a new type of vertebrae probe.  Other than the DXA scan, we are not aware of any viable alternatives.  Our transmission probe concept is ideal for this application, but key building blocks need to be demonstrated before progressing to developing a surgically compatible device – recovering the dielectric properties from the probe measurements and developing a suitable technique for measuring the separation distance between the probe tips.  We have proved the feasibility of this technique in this paper.  Finally, we have previously described the limitations of the reflection-based microwave dielectric probes.  We have added a few sentences describing the limitations of an optical probe in this context.  Other than that, we are unaware of related approaches for this application.

- Figure 1 does not respect the correct formatting according to the MDPI format.

In reviewing the figure requirements in the MDPI instructions, we are not aware that the formatting is incorrect.  We have used the template provided by MDPI.  This would most likely not be a large change if necessary, but we will defer to the SENSORS editorial board for further instructions.

  1. Materials and Methods

- Figures 2-5 do not respect the correct formatting according to the MDPI format.

See Response for the MDPI formatting requirements for Figure 1.

- Equations should be processed so that notations and indices can be easily identified.

We have increased the size of the font for the equations.  The description for each term is included in the text immediately following the equation for which the terms are introduced.  We have added a reference to a well-regarded textbook that can be referred to if the reader requires more information.

  1. Results

- Figure 7 does not respect the correct formatting according to the MDPI format.

See Response for the MDPI formatting requirements for Figure 1.

- Figures 9, 10, and 11 should be redone because they seem unclear, and the associated legends are hard to read.

We have increased the size of all fonts.  We have also increased the sizes of the line widths.  Especially for Figures 8 and 9 (previously Figures 9 and 10), we have grouped all of the curves for the different concentrations by color along with the actual values (which are much wider than the others).  We think this makes these three figures more readable.

- Figures 12 and 13 do not respect the correct formatting according to the MDPI format.

See Response for the MDPI formatting requirements for Figure 1.

  1. Discussion

In section 4, the results need to be discussed and considered in other studies. Therefore, please refer to these studies. Also, it would be best to describe your research's limitations.

To the best of our knowledge, we are unaware of any studies looking into alternative approaches for assessing the vertebrae strength other than DXA scans.  It is a unique research topic and open to other competing solutions.

With respect to limitations, the Discussion section already brings up important points that still need to be addressed in the development of these probes.  The primary challenges at this point are:  (1) making the probes stiffer to reduce the position accuracy, (2) sterilizability of the probes, (3) compatibility of the probes with human tissue, and (4) how this will fit into the work flow of the actual surgery.  At present these are essentially the chief limitations.  As alluded to, these are all topics for further research and development.

Conclusion

Here, in addition to conclusions regarding the obtained results, you must also present the following:

- What are the main challenges of the current work?

We have substantially added to the Conclusions section which is now called the Conclusions and Future Works section.  In particular, we have highlighted the main challenges of getting this device into the clinic.  Of primary importance will be workflow and sterilization.  We have described the trade-offs which add to the importance of having performed the analyses on the surgical navigation-based concept along with the machine vision concept in this manuscript.

References

References should be more extensive and up-to-date. At least 70% must be from the last 10 years.

We are unaware that there is a requirement for 70% of the references to be within the last 10 years.  As requested by other reviewers, we have added substantially to the number of references.  In our case, more than 50% were within the past 10 years for the initial submission.  Some of the older references have been strategically chosen because they were seminal for that particular topic.  In the cases where we have selected textbooks, these are considered seminal in the microwave field are all currently used in college classes and quite relevant to the associated topic.  While the use of the open-ended coaxial probes is quite novel, they do tap into traditional microwave engineering practices as noted by these references.  I should note that in the revised manuscript, 26 of the 40 references are within the last 10 years – 65%.  However, there are several that are quite close – 4 extra within 11 years.  If you included those, 75% are within the last 11 years.

Reviewer 5 Report

The paper has a new application of transmission probe. The basic theory is solid. The corresponding expansion based on the basic theory is developed well. The experimental method and the results are presented well. My suggestions are as follows:

(1) The structure of the thin coaxis probes should be introduced since they are critical for sensing data.

(2) QR code usage in the application decides the accuracy of the system, so besides the calibration of the camera, please show the accumulated error by using the QR code.

(3) The specific application, in equations 1 and 2, the corresponding constants and parameters (relative permittivity, conductivity, radian frequency, free 244 space magnetic permeability and the free space permittivity). Can you explain why the conventional values of these are still fit the specific application?

Author Response

Dear reviewer,

Thank you for your thoughtful and constructive criticisms.  Below we answer the questions raised and corresponding changes have been made to the revised text.

The paper has a new application of transmission probe. The basic theory is solid. The corresponding expansion based on the basic theory is developed well. The experimental method and the results are presented well.

Thank you.

My suggestions are as follows:

(1) The structure of the thin coax is probes should be introduced since they are critical for sensing data.

Several sentences more thoroughly describing the probes have been added to Section 2.2.

(2) QR code usage in the application decides the accuracy of the system, so besides the calibration of the camera, please show the accumulated error by using the QR code.

We provided a thorough quantitative analysis of the overall error using this machine vision approach and compared it against that for what would be a viable technique, i.e. surgical navigation.  Separating out the errors for each contributor within the processes is beyond the scope of the current manuscript but could be interesting to the readership in the future.

(3) The specific application, in equations 1 and 2, the corresponding constants and parameters (relative permittivity, conductivity, radian frequency, free space magnetic permeability and the free space permittivity). Can you explain why the conventional values of these are still fit the specific application?

With respect to the microwave signal propagation governed by Maxwell’s equations, the dielectric properties govern the behavior [Pozar – textbook].  These properties are the electrical permittivity, electrical conductivity, and the magnetic permeability.  In all practical situations we will encounter, there will not be any magnetic material.  As such, the permeability does not come into play in our derivation of the Maxwell’s equations and it can be assumed to be that of free space for the entire experimental domain.  In our case, we are primarily interested in the permittivity and conductivity.  It is common convention to treat the permittivity as the product of the free space eo permittivity and the relative permittivity, er (a unitless quantity). 

For the governing Maxwell’s equation, the radian frequency is not a property of the medium.  It is a feature of the applied fields.  In this case, the radian frequency is equal to 2pf where f is the frequency of the electromagnetic wave.  We have made note of this and added a reference to the text.

Round 2

Reviewer 1 Report

The authors have accepted all my recommendations. I agree with  the paper publishing. Thank you.⁹

Reviewer 4 Report

Thank you for your answers. Apart from the observation that some figures could be clearer, I don't have any other comments.